# Laser Velocimetry for the In Situ Sensing of Deep-Sea Hydrothermal Flow Velocity

**DOI:** 10.3390/s23208411

**Published:** 2023-10-12

**Authors:** Jingjing Sun, Lei Zhang, Guojie Tu, Shenglai Zhen, Zhigang Cao, Guosheng Zhang, Benli Yu

**Affiliations:** Information Materials and Intelligent Sensing Laboratory of Anhui Province, Anhui University, Hefei 230601, China; sunjingjing@ahu.edu.cn (J.S.); optzl@ahu.edu.cn (L.Z.); 15041@ahu.edu.cn (G.T.); slzhen@ahu.edu.cn (S.Z.); caozhigang@ahu.edu.cn (Z.C.); 22752@ahu.edu.cn (G.Z.)

**Keywords:** laser velocimetry, ocean exploration, in situ sensing, flow velocity

## Abstract

Laser Doppler velocimetry (LDV) based on a differential laser Doppler system has been widely used in fluid mechanics to measure particle velocity. However, the two outgoing lights must intersect strictly at the measurement position. In cross-interface applications, due to interface effects, two beams of light become easily disjointed. To address the issue, we present a laser velocimeter in a coaxial arrangement consisting of the following components: a single-frequency laser (wavelength λ = 532 nm) and a Twyman–Green interferometer. In contrast to previous LDV systems, a laser velocimeter based on the Twyman–Green interferometer has the advantage of realizing cross-interface measurement. At the same time, the sensitive direction of the instrument can be changed according to the direction of the measured speed. We have developed a 4000 m level laser hydrothermal flow velocity measurement prototype suitable for deep-sea in situ measurement. The system underwent a withstand voltage test at the Qingdao Deep Sea Base, and the signal obtained was normal under a high pressure of 40 MPa. The velocity contrast measurement was carried out at the China Institute of Water Resources and Hydropower Research. The maximum relative error of the measurement was 8.82% when compared with the acoustic Doppler velocimeter at the low-speed range of 0.1–1 m/s. The maximum relative error of the measurement was 1.98% when compared with the nozzle standard velocity system at the high-speed range of 1–7 m/s. Finally, the prototype system was successfully evaluated in the shallow sea in Lingshui, Hainan, with it demonstrating great potential for the in situ measurement of fluid velocity at marine hydrothermal vents.

## 1. Introduction

Submarine hydrothermal vents continuously erupt magma and high-temperature hydrothermal fluid into the sea through a unique chimney-shaped vent, which is an important influencing factor on the Earth’s heat flux and chemical element cycle. Its quantitative calculation relies on the accurate measurement of multiple physical parameters of the hydrothermal system, such as the scale of the hydrothermal zone, the density of the hydrothermal fluid, the specific heat capacity at constant pressure, the temperature of the hydrothermal fluid, and the flow rate of the hydrothermal fluid. If the hydrothermal velocity measurement can be realized, it can provide an effective test method for the material flux analysis of the submarine hydrothermal output and provide an important reference for the survey of hydrothermal ore-forming conditions and ore-forming rate.

Nowadays, intrusive devices such as vane flowmeters and turbine flowmeters are commonly used to estimate flow rate. However, the measuring equipment easily interferes with the flow rate of the point to be measured. Laser Doppler velocimetry (LDV) is a noninvasive technique for measuring the speed of moving solid-state surfaces or particle velocity in a fluid by laser interferometry [1,2,3,4,5,6,7,8,9,10,11,12,13,14]. Researchers continuously develop and design LDV devices suitable for different applications [15,16,17,18,19]. LDV devices with a dual-beam structure have been widely used for fluid flow measurements for many years [20,21,22,23,24,25,26,27]. This scheme requires the two beams to intersect strictly to form interference fringes. However, with the increase in seawater pressure, the deformation of the light window of the pressure cabin makes it difficult for the double beams to keep crossing in hydrothermal velocity detection.

A Twyman–Green interferometer is one in which a division of amplitude takes place [28,29,30]. Interference occurs from the recombination of two parts of a beam of light, which is divided by partial reflection and partial transmission at a beam-splitter. The two beams of light follow different paths and are reflected to the beam splitter from mirrors. Two beams of reflected light interfere and focus on the measurement position through the lens. And it is the position of the mirrors which determines the nature of the interference. The method is therefore expected to perform well in eliminating the interference of light window deformation for velocity measurement.

The previously reported Twyman–Green interferometers are used to test optical components such as lenses and prisms. The paper describes the system and experiments to measure the speed of an optical chopper.

## 2. Detection Principle

The optical layout of the system is shown in Figure 1. A fiber-optic laser (532 nm) is used as the light source. The laser beam passes through the beam expander and splits into two beams after it passes through the beam splitter. Mirrors M1 and M2 reflect the two beams to the beam splitter. Both mirror M1 and M2 can be tilted. A virtual wedge is formed between M1 and M2. When the path lengths for the mirrors are nearly equal, the wedge causes the reflected wavefronts to coincide with each other such that fringes are formed. Almost all light with interference fringes passes through a polarization beam splitter (PBS) and quarter-wave plate; then, lens L1 focuses the beam onto the rotating side of the optical chopper through the light window. The side of the optical chopper is where the measured speed is located. The arrow in Figure 1 indicates the direction of rotation of the optical chopper. The backscattered light is collected by lens L1, reflected by a polarization beam splitter, and focused by lens L2 into the photodetector (PD).

For this system, it is necessary to take a magnified view of the fringes to determine the characteristics of the interference fringes. Therefore, a charge-coupled device (CCD) camera connected to a monitor is implemented to observe the fringes through lens L3, using a small amount of reflected light from the PBS. The spot on the target is the same size as the spot on the CCD. To accurately position the laser spot on the optical chopper and the CCD, both the optical chopper and the CCD are installed on a three-dimensional micro-motion platform whose linear adjustment resolution is 1 mm. We can adjust the movement of the two optical platforms at the same time to measure the speed and the fringe interval. The spacing of the interference fringes is calculated jointly by the CCD and computer.

The last step is to obtain suitable fringes at the focal point of L1 with both mirrors exposed. It may be necessary to realign M1 until the spots coincide. A CCD placed at the focal point of L3 should see some fringes. The position of M1 can then be adjusted a small amount until the greatest contrast is obtained. The path lengths of the two interferometer arms are equal. Adjustments can be made to M2 so that the fringe spacing is increased or decreased. And the direction of the fringes can also be changed. The fringe spacing d of interference light reaching the PBS depends on the wavelength λ of the laser light and the angle ϕ between both mirrors.
(1)$$d=\frac{\lambda }{2\phi }$$

Interference fringes are focused on the side of the optical chopper and the CCD through lenses L1 and L3, respectively. The fringe spacing is denoted as d1. The particle of the moving surface scatters light while crossing the fringes. The speed direction should be perpendicular to the fringe direction. The light scattered is then detected by a photodetector and its intensity is modulated with a frequency equal to:(2)$$f_{D}=\frac{Vcos\theta }{d1}$$
where V  is the velocity vector on the side of the optical chopper. *θ* is the angle between the direction of velocity and the direction perpendicular to the stripe. Then, $f_{D}$ determination allows velocity estimation. It can be obtained by the Fourier transform of the detection signal. Its spectrum consists of two parts. One part is the spectrum of the low-frequency base signal. The other part is the signal spectrum. The base spectrum can be filtered by a high pass filter and the signal containing speed information can be reserved for processing. The ${\;f}_{D}$ obtained from the output signal of the photodetector and d1 calculated by the CCD can be used to calculate Vcosθ using the transformation formula of Equation (2). Try to make θ equal to 0, and at this point, the velocity measured by the laser velocimeter is the velocity perpendicular to the stripe. In general, the value of θ cannot be determined. When the measured velocity is not able to be perpendicular to the interference fringes, measurement errors will be introduced.

## 3. Experiments and Results

This section provides a concise and precise description of the experimental results and their interpretation, as well as the experimental conclusions.

The transient velocity of the optical chopper was next tested to demonstrate the working principles of the probe. The optical chopper is a precision instrument with a stable rotation speed and less jitter. The rotation speed was allowed to vary between 25 and 500 revolutions per minute. The fringe direction was adjusted to be perpendicular to the linear velocity of the optical chopper surface. The fringe spacing observed using the CCD was 6.4 um. The number of fringes was 27. The experiments were designed to measure the changes in frequency $f_{D}\;$ with the change in rotating speed. With the frequency and fringe spacing, the speed obtained by the laser velocimeter is shown in Figure 2. Each value is an average of three replicates. For comparison, theoretical speed was also calculated with the rotation diameter and rotation speed displayed by the chopper. The speed results match very well.

The results show that the speed measured by the laser velocimeter increased with the increasing rotation speed. It is important to note that the observed trend is linear. It means that the increase in the speed was due to the increase in the rotation speed. The speed measured by the laser velocimeter could reflect changes in rotation speed without distortion. The result is in agreement with the dual-beam LDV. Both of these speed measurement methods are based on the generated fringes, and how the fringes are generated is different. Therefore, the method proposed can achieve speed measurement and is expected to have more advantages in certain aspects.

When the rotational speed of the chopper was determined to be 100 revolutions per minute, the signal-to-noise ratio of the spectrum of the PD output signal was as shown in Figure 3. The number of fringes can be changed by adjusting the mirrors. As the number of fringes increases, the signal-to-noise ratio first increases and then decreases. When the number of fringes is less than 10 and greater than 80, the signal-to-noise ratio is relatively low. When the number of interference fringes is between 20 and 45, the signal-to-noise ratio is relatively good.

The results show that the signal-to-noise ratio is affected by the number of interference fringes. Too many or too few interference fringes cannot achieve a good signal-to-noise ratio, which affects the accuracy of the final measurement speed. Therefore, the appropriate number of fringes is a factor to consider when designing a speed measurement system.

When the same spot size is used, different fringe numbers indicate different fringe intervals. The trend of signal-to-noise ratio with increasing speed at different fringe intervals is shown in Figure 4. Regardless of the fringe spacing, the signal-to-noise ratio tends to decrease with increasing speed. When the fringe spacing is appropriate, the overall signal-to-noise ratio is higher. When the fringe spacing is 2.8 um and 4.0 um, the signal-to-noise ratio of the signal is higher. This conclusion is consistent with the conclusion in Figure 3. Of course, the specific fringe spacing may need to be determined based on the object being tested.

The results indicate that the signal-to-noise ratio is affected by the magnitude of the measured speed. As the signal speed increases, the signal may deteriorate, making it difficult to extract effective speed information. At this point, it may be necessary to use methods such as increasing optical power or signal processing to improve. 

To simulate the effect of the deformation of light windows under different water pressures on signal pickup, different shaped light windows were made and placed on the outgoing light path. Assuming that the water depth increases from 0 to 5000 m and for every 1000 m of water depth increase, corresponding light windows 1–6 are made based on the deformation of the light window. When changing the rotation speed, the speed obtained by the system under different light windows is shown in Figure 5. The shape variables of light windows 1 to 6 increase sequentially. It can be seen that this laser velocimetry optical path can resist the influence of changes in the shape of the optical window on signal pickup. The significant overlaps indicate that the deformation of the optical window has a relatively small impact on the acquisition of speed signals. It seems that the proposed interferometer scheme removes the problems associated with signal distortion by the light porthole at depth.

In an effort to apply this system to the measurement of flow velocity at deep-sea hydrothermal vents, the laser interference velocity measurement system was improved and packaged using a pressure chamber. A charged pressure test was conducted at the Qingdao Deepsea Base Management Center, with the nozzle flow velocity of the hydrothermal velocity simulation device as the testing target. The maximum test pressure is 40 MPa, the pressure was maintained for 20 min, and the system signal acquisition was normal.

To verify the accuracy of the laser velocimeter (LV) in measuring the water flow velocity, an acoustic Doppler velocimeter (ADV) was used to conduct a comparison test on the velocity of 0.1–1 m/s at the China Institute of Water Resources and Hydropower Research. The flow velocity measurement range of the ADV is 0–3 m/s, with a measurement accuracy of ±0.5% and a sampling frequency of 100 Hz. During testing, the ADV was fixed in the water tank which can generate a relatively stable water flow. The LV was installed on one side of the water tank, and the laser emitted by the system was injected into the water through the organic glass side wall of the water tank. We set 11 speed nodes and after the flow rate stabilized, we simultaneously sampled and measured the flow rate. Each sampling time was 3 min, and the test results are shown in Figure 6. The maximum relative measurement error between the laser velocity measurement system and the ultrasonic Doppler velocity meter is −8.82%. The reason for the error is that the stability of the water flow in the sink is relative, and the flow velocity measured by the ADV is the average velocity within the measured volume (*L*10 × ∅15 mm), while the laser velocimetry system is a point measurement with a spatial resolution of several tens of micrometers.

The nozzle method was used to compare and test the flow velocity in the high-speed range of 1–7 m/s. The special device of the system ensures the uniform distribution of the nozzle flow velocity. During the test, the instrument was installed above the nozzle, and the laser emitted by the system was focused on the water. The nozzle standard flow rate system was set with 11 speed nodes, and the laser velocity measurement system measured the flow rate at the nozzle. At present, the maximum flow rate of this flow rate calibration device is 302 L/s, equivalent to 9.7 m/s. The speed value set by the nozzle standard flow rate system was compared with the data measured by the laser velocity measurement system. The experimental results show that the maximum measurement error of the system in the high-speed section is 1.98%, as shown in Figure 7. The average relative measurement error in this flow rate range is small, and the flow rate at the nozzle of the nozzle standard flow rate system is stable. 

Based on the above experimental results, a shallow sea experiment was conducted on the laser velocity measurement system in Lingshui, Hainan. The crane lowered the system to a depth of 50 m underwater, and the system measured the descent speed during this process. After hanging the laser velocimeter at the stern of the ship, the towing speed of the ship was measured and compared with the velocity values measured by the acoustic Doppler velocimetry system. The installation of the equipment is shown in Figure 8, and the results show that the max. relative error of the towing speed tested by the two is 7.3%. 

It should be noted that when testing was completed in stable water flow, the relative error was relatively small. This indicates that the speed measurement results of the laser velocimeter are relatively reliable. Of course, if the comparative experiment is completed using a single point dual beam laser Doppler velocimeter (LDV), the results will be more direct. Further experiments will be carried out when such conditions are met.

## 4. Conclusions

The proposed approach has given a novel route for the non-contact measurement of speed based on a Twyman–Green interferometer. It implements a coaxial architecture of two outgoing lights and can be used for cross-interface measurement applications. The direction and spacing of the springs can be changed by adjusting the mirror M2. Thus, the speed of different directions and different particle sizes can be measured. 

A prototype of the laser flow velocity measurement principle was built to meet the requirements of the in situ measurement of deep-sea hydrothermal flow velocity, and pressure resistance tests and flow velocity comparison tests were conducted. The prototype of this principle is 500 mm in length and 205 mm in diameter. It is powered and communicates with the outside world through a watertight wire. Considering the low loss of blue–green light caused by seawater, a narrow linewidth green laser with a wavelength of 532 nm is used internally.

The pressure withstand test was carried out at the Qingdao Deep Sea Base with the prototype, and the speed contrast test was carried out at the China Institute of Water Resources and Hydropower Research. The measurement error in the main flow velocity section was less than 5%. A shallow sea test was conducted in Lingshui, Hainan. A crane lowered the system to a depth of 50 m underwater to test the descent speed of the system. The signal was obtained well. The test prototype and acoustic Doppler velocimeter were hung together at the stern of the ship, making it to 2 m underwater to test the towing speed with the ship. The experimental results show that the max. relative error of the drag speed between the two tests is 7.3%. The experimental results demonstrate the feasibility of using a laser velocimeter based on a Twyman-Green interferometer for the in situ measurement of deep-sea hydrothermal flow velocity.

## Figures and Tables

**Figure 1 sensors-23-08411-f001:**
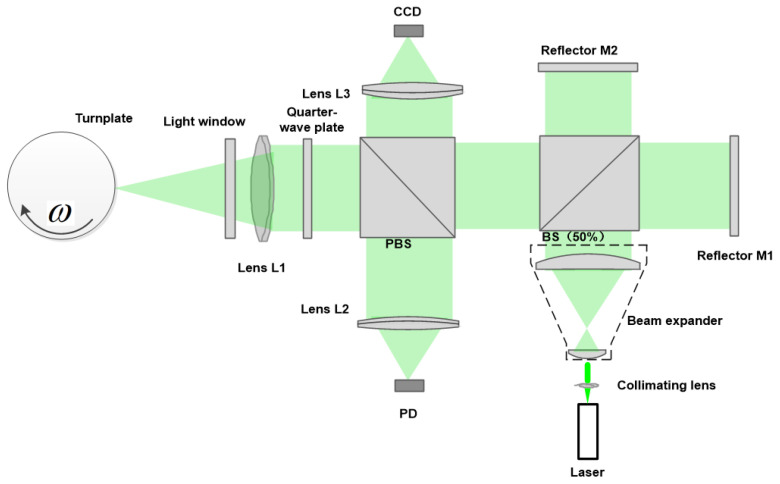
Optical layout of an Twyman–Green interferometer.

**Figure 2 sensors-23-08411-f002:**
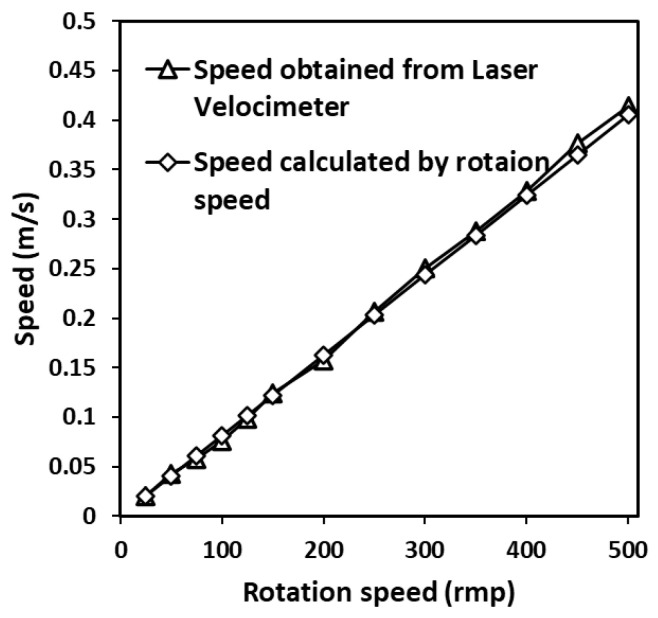
The speed obtained by the laser velocimeter and the speed calculated by the rotational speed.

**Figure 3 sensors-23-08411-f003:**
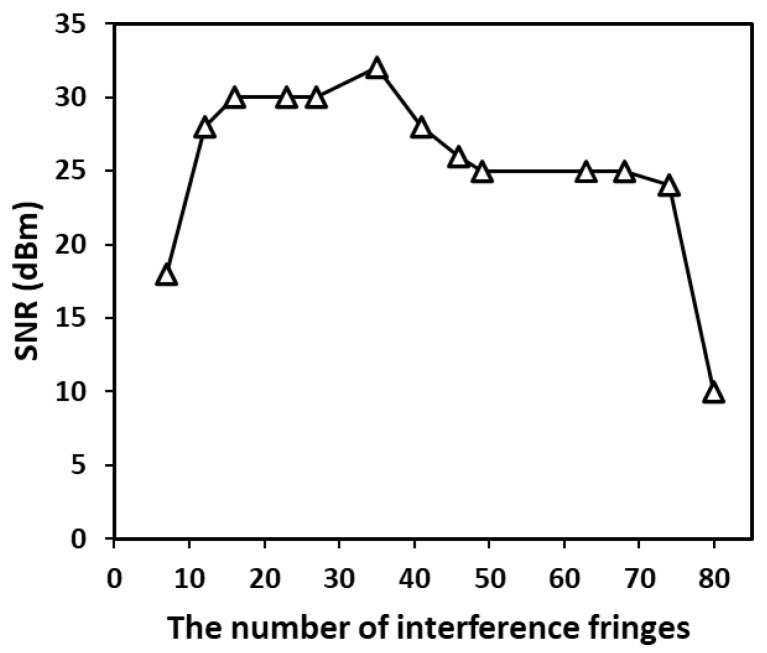
The signal-to-noise ratio with a different number of fringes.

**Figure 4 sensors-23-08411-f004:**
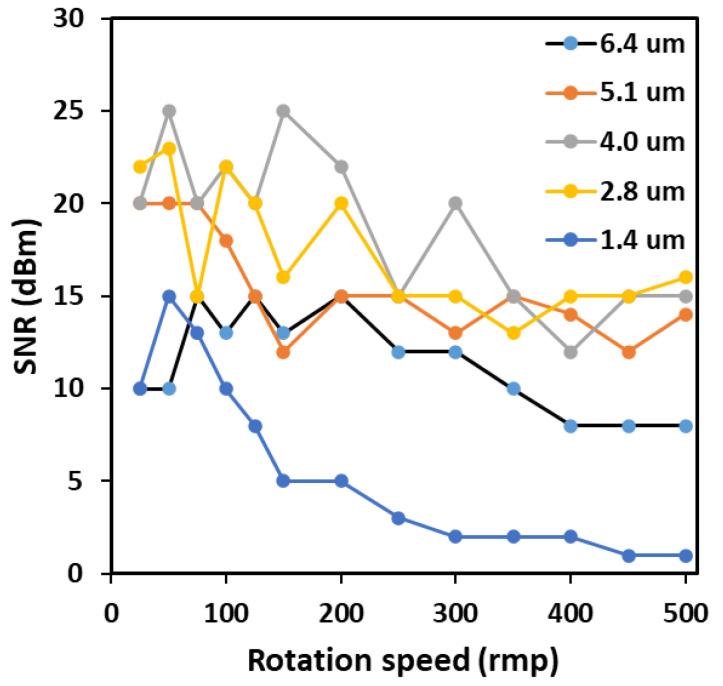
The variation of signal-to-noise ratio with rotational speed at different fringe intervals.

**Figure 5 sensors-23-08411-f005:**
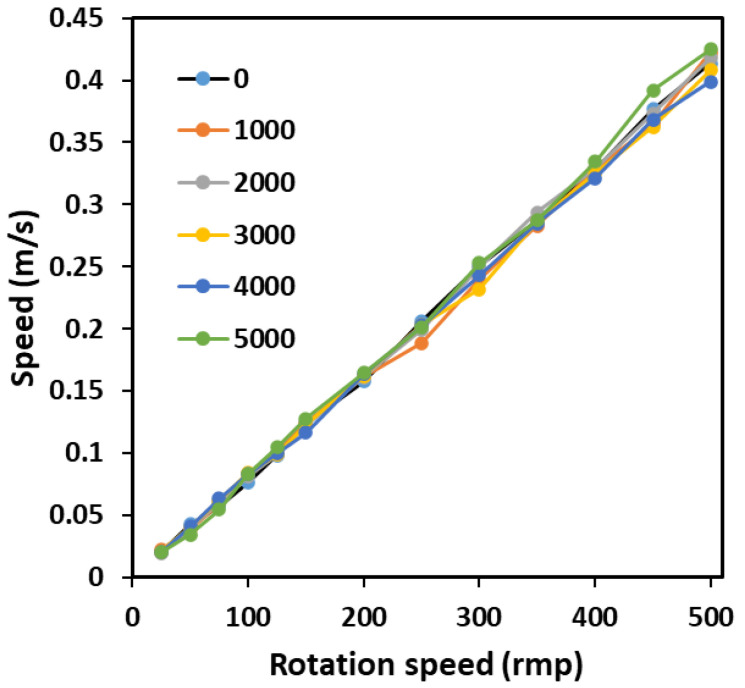
The speed obtained by the laser velocimeter using different light windows.

**Figure 6 sensors-23-08411-f006:**
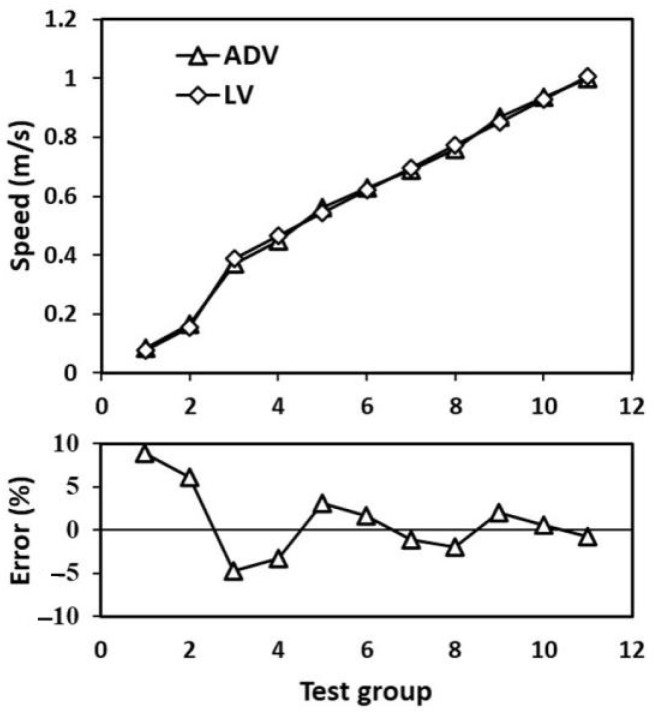
Comparison of the laser velocimetry system and the acoustic Doppler velocimeter.

**Figure 7 sensors-23-08411-f007:**
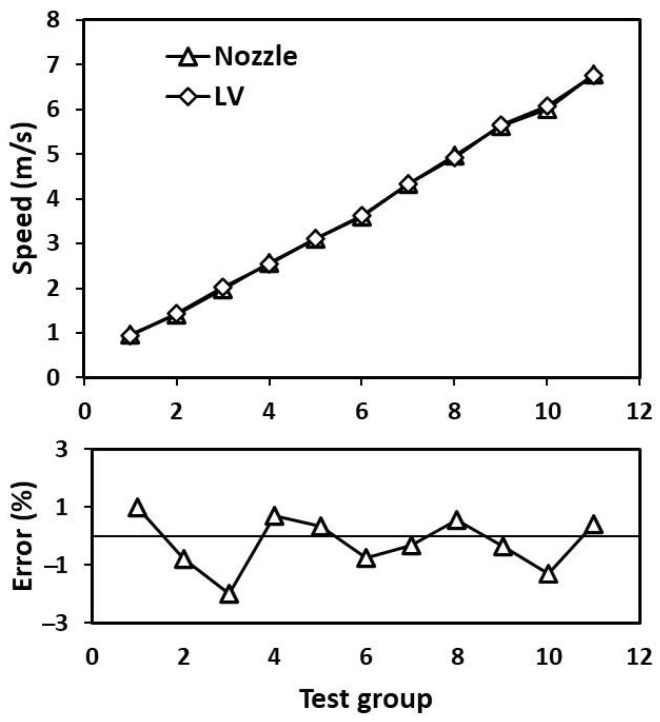
Comparison test of the laser velocimetry system and the standard nozzle flow system.

**Figure 8 sensors-23-08411-f008:**
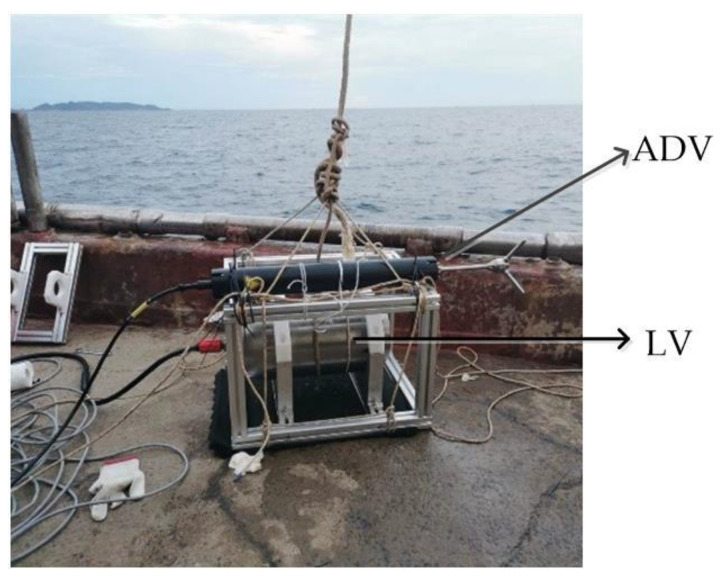
The installation of the equipment of the LV and the ADV in the shallow sea test.

## Data Availability

Not applicable.

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
