# Peer review of "Laser Velocimetry for the In Situ Sensing of Deep-Sea Hydrothermal Flow Velocity"

_sensors, 2023, doi:10.3390/s23208411_

Round 1
Reviewer 1 Report
1.There are few references in recent years.
2.In Figure 3, Code 1-6 can be canceled and replaced by 0-5000.
3.For figure 4a, it is not clear, and Does it make any sense?
4.The fonts in all the figures are not uniform and of low quality.
5.There is a lack of discussion and specific analysis of the data.
6.The project is 2016 year?
7.It is recommended that the author organize the data well, beautify the graphs, and mine and analyze the data well.
Modify Chinese expressions and grammatical errors in the text.
Reviewer 2 Report
|
The development of new sensors for noncontact measurement of flow velocity is relevant, especially for measurements at depth. However, this article does not give a complete picture of the method of measurement on the sensor, which was developed by the authors. The article needs to be improved. First of all, in terms of describing the principle of measuring the fluid flow velocity using a two-beam Twyman-Green interferometer. Figure 1 should give the reader an idea of where the measured fluid flow is located and how the interference pattern parameters are measured and the subsequent flow velocity is measured. This will make it possible to understand the novelty of the proposed sensor and its difference from the already known and used LDVs. On pages 98 – 104, the authors give only general considerations on how the flow rate is measured (estimated). Formula (2) does not clarify how the speed is calculated. The value "ν is the magnitude of the component of the velocity vector ? perpendicular to the fringes" is absent from formula (2) at all. It is necessary to clarify the statement that the use of the proposed interferometer scheme removes the problems associated with signal distortion by the light porthole at depth. This was one of the reasons for choosing the Twyman-Green scheme. There is a note to the name of the interferometer. The selected type of interferometer refers to the type of two-beam interferometers, and if we follow the historical terminology, then the use of two flat mirrors (M1 and M2) rather corresponds to the Michelson interferometer. There is no diagram, photo and detailed description of the sensor prototype itself, which was tested from a ship in shallow water. The authors compare the results of the flow rate experiment with those obtained using the Acoustic Doppler Velocimeter (ADV). At the same time, it would be logical to make a comparison with well-known Laser Doppler Velocimetry (LDV) devices, at least in laboratory conditions. |
Reviewer 3 Report
This is an interesting, but very incomplete article. There are a lot of inaccuracies and flaws in the article.
1. In Figure 1 there is no description in the notation caption. From the explanation it is not clear why a module consisting of a divider and two mirrors M1 and M2 is needed.
2. After reflection from the M1 and M2 mirrors, the beam falls back into the laser, which leads to the departure of the laser stability. How do you deal with this?
3. It is not described what is the sensitive element.
4. Why is an SF camera and a photo detector used?
5. What is the advantage of this laser velosimeter over others, for example, the ProSpeed LSV-2100
I believe that the article is suitable for the subject of the magazine, but requires a lot of refinement.
English needs to be corrected, it is difficult to understand. Some proposals are not coordinated, there are errors.
Round 2
Reviewer 1 Report
The data is sparse and not rich. Questions need to be refined and comparable in order to highlight their unique characteristics and technical features or advancement.
It is ok.
Author Response
The data is sparse and not rich. Questions need to be refined and comparable in order to highlight their unique characteristics and technical features or advancement.
Response and reply: Thank you for this suggestion. We have added Figures 3 and 4 and their analysis in the article. The related description has been provided in the revised paper.
Reviewer 2 Report
The article may be published in present form
Author Response
The article may be published in present form.
Response and reply: Thank you for your valuable feedback.
Reviewer 3 Report
The manuscript can be published in this form
Author Response
The manuscript can be published in this form.
Response and reply: Thank you for your valuable feedback.